# Functional Role of AGAP2/PIKE-A in Fcγ Receptor-Mediated Phagocytosis

**DOI:** 10.3390/cells12010072

**Published:** 2022-12-24

**Authors:** François C. Chouinard, Lynn Davis, Caroline Gilbert, Sylvain G. Bourgoin

**Affiliations:** 1Centre de Recherche du CHU de Québec—Université Laval, Québec City, QC G1V 4G2, Canada; 2Département de Microbiologie-Infectiologie et D’immunologie, Université Laval, Québec City, QC G1V 0A6, Canada; 3Centre ARThrite de L’université Laval, Québec City, QC G1V 4G2, Canada

**Keywords:** ArfGAP, FcγR, phagocytosis, ADP-ribosylation factor, neutrophils, PLB-985 cells

## Abstract

In phagocytes, cytoskeletal and membrane remodeling is finely regulated at the phagocytic cup. Various smaFll G proteins, including those of the Arf family, control these dynamic processes. Human neutrophils express AGAP2, an Arf GTPase activating protein (ArfGAP) that regulates endosomal trafficking and focal adhesion remodeling. We first examined the impact of AGAP2 on phagocytosis in CHO cells stably expressing the FcγRIIA receptor (CHO-IIA). In unstimulated CHO-IIA cells, AGAP2 only partially co-localized with cytoskeletal elements and intracellular compartments. In CHO-IIA cells, AGAP2 transiently accumulated at actin-rich phagocytic cups and increased Fcγ receptor-mediated phagocytosis. Enhanced phagocytosis was not dependent on the N-terminal GTP-binding protein-like (GLD) domain of AGAP2. AGAP2 deleted of its GTPase-activating protein (GAP) domain was not recruited to phagocytic cups and did not enhance the engulfment of IgG-opsonized beads. However, the GAP-deficient [R618K]AGAP2 transiently localized at the phagocytic cups and enhanced phagocytosis. In PLB-985 cells differentiated towards a neutrophil-like phenotype, silencing of AGAP2 reduced phagocytosis of opsonized zymosan. In human neutrophils, opsonized zymosan or monosodium urate crystals induced AGAP2 phosphorylation. The data indicate that particulate agonists induce AGAP2 phosphorylation in neutrophils. This study highlights the role of AGAP2 and its GAP domain but not GAP activity in FcγR-dependent uptake of opsonized particles.

## 1. Introduction

Small GTPases of the Arf (ADP-ribosylation factor) family are converted to the active GTP-bound form by guanine nucleotide exchange factors (GEFs) that displace GDP to allow GTP binding [1], and GTPase-activating proteins (GAPs) have the opposite effect through stimulation of GTP hydrolysis [2]. Thirty-one predicted human genes encoding proteins with Arf GAP domains are classified in ten subfamilies, based on domain structure and phylogenetic analyses [3]. The AZAP family includes the ASAP, ACAP, ARAP and AGAP subfamilies sharing an Arf GAP domain sandwiched between pleckstrin homology (PH) and ankyrin (ANK) repeat domains [4]. The three AGAP proteins differ from other AZAP family members by an additional N-terminal GTP-binding protein-like domain (GLD) and a split PH domain [2]. AGAP2 [5], also named phosphoinositide 3-kinase enhancer (PIKE)-A [6] and GGAP2 [7] by other groups, was initially identified as a full-length cDNA designated KIAA0167 in human immature myeloid cell line KG-1 [8]. A single gene gives rise to three protein isoforms named PIKE-L, AGAP2, and PIKE-S, through the use of different start codons and alternative splicing. The domain structure of PIKE-L, the long splice variant, resembles that of AGAP2 with a N-terminal proline-rich extension (PRD) [9]. The shorter isoform PIKE-S shares its N-terminal with PIKE-L, as well as the GLD and a part of the PH domain, but lacks Arf GAP and ANK repeat domains [10]. The expression of PIKE-S and -L, but not that of AGAP2, is restricted to brain [7,8,9,10,11,12,13]. The N-terminal PRD of nuclear PIKE-S and cytosolic PIKE-L interacts directly with phosphatidylinositol 3-kinase (PI3K) [9,10]. This interaction stimulates PI3K and the protein kinase Akt, leading to prevention of neuronal apoptosis [9,14,15]. AGAP2, which lacks the N-terminal PRD, does not bind to PI3K. Instead, AGAP2 interacts directly with and is phosphorylated by the downstream effector of PI3K, Akt [16]. Interaction between AGAP2 and active Akt further enhances Akt activity leading to human cancer cell survival and promoting cancer cell invasion and migration [17]. AGAP2 phosphorylation by kinases such as Fyn, AMP-K and cyclin dependent kinase 5 contributes to regulation of cancer progression [18,19,20]. Apart from its compelling role in oncogenesis, AGAP2 may regulate membrane trafficking by affecting the AP-1/Rab4 endosomal compartment [2].

In macrophages, as well as in Chinese hamster ovary cells expressing human FcγRIIA (CHOIIA), several intracellular compartments contribute to the formation of nascent phagosomes, including recycling endosomes [21,22,23,24] late endosomes and lysosomes [25,26]. Arf6 has been reported to deliver vesicles from the recycling compartments to the forming phagosome [23], and to control cytoskeletal reorganization during early events of phagocytosis [27,28,29]. Arf1 may also contribute indirectly to the process of phagocytosis [30,31] and be involved in the recruitment of AP-1 recycling endosomes at early stages of phagocytosis [32]. Although FcγR-mediated phagocytosis has been studied extensively, only one ArfGAP, named ASAP2/PAG3/KIAA0400, has been reported to affect FcγR-mediated phagocytosis through regulation of Arf6 activity [28].

In this study we demonstrate that human neutrophils express AGAP2 but not AGAP1. Using CHO-IIA and mouse macrophages, we found that AGAP2 was recruited to the forming phagocytic cups but not to fully internalized phagosomes. AGAP2 enhanced FcγR-mediated phagocytosis in CHO-IIA independently of its GAP activity. In contrast, silencing of AGAP2 in the neutrophil-like cell line PLB-985 reduced phagocytosis. Furthermore, particulate agonists induce AGAP2 phosphorylation in human neutrophils. Taken together, our data suggest that AGAP2 positively regulates FcγR-dependent uptake of opsonized particles by phagocytic cells.

## 2. Materials and Methods

### 2.1. Reagents

Ficoll-Paque, Mg^2+^-free Hank’s balanced salt solution (HBSS), Tris-HCl, HEPES, PBS, BSA, fetal bovine serum (FBS), donkey serum, RPMI-1640, DMEM, α-MEM, L-glutamine, l-cysteine, glycine and G-418 were obtained from Wisent (St-Bruno, QC, Canada). Sodium orthovanadate (Na_3_VO_4_), PMSF, pNPP, soybean trypsin inhibitor, Triton X-100, human IgGs, Zymosan-A BioParticles (Z4250-1G), N-formyl-Methionyl-Leucyl-Phenylalanine (fMLF) and dibutyryl cyclic AMP (dbcAMP) were purchased from Millipore-Sigma (Oakville, ON, Canada). Dextran T-500 was obtained from GE Healthcare (Baie d’Urfé, QC, Canada). Aprotinin and leupeptin were obtained from Roche Applied Science (Laval, QC, Canada). Diisopropyl fluoro-phosphate (DFP) was purchased from Serva Electrophoresis (Heidelberg, Germany). Calcein acetoxymethylester (calcein-AM) and NP-40 were from Calbiochem (Gibbstown, NJ, USA). Zymosan A (*S*. *cerevisiae*) BioParticles Alexa Fluor 488 and 594 conjugates and ProLong Gold antifade reagent were obtained from Invitrogen (Burlington, ON, Canada). Polystyrene (2.93 µm) and cross-linked polystyrene/2% divinylbenzene (3.87 µm) beads were from Bangs Laboratories (Fishers, IN, USA). Trypan Blue was purchased from EM Diagnostic Systems (Gibbstown, NJ, USA). Paraformaldehyde (PFA) was obtained from Laboratoire MAT (Québec, QC, Canada). [^32^P]orthophosphate (1000 Ci/mmol) was purchased from PerkinElmer (Norwalk, CT, USA). Triclinical MSU monohydrate crystals were prepared as described previously [33].

### 2.2. Antibodies

Polyclonal AGAP2 antibody was produced in rabbits using a His-tagged N-terminal domain of AGAP2 (amino acids 131–279) as immunogen. Polyclonal antibody CT10 is an IgG-purified fraction of a rabbit antiserum against the cytoplasmic domain of FcγRIIA [34]. Actin monoclonal antibody was obtained from Sigma-Aldrich (Oakville, ON, Canada). Anti-phosphotyrosine (4G10) was from Upstate Biotechnology (Lake Placid, NY, USA). Texas Red-conjugated donkey anti-human IgG, HRP-labeled donkey anti-rabbit and sheep anti-mouse IgG were from Jackson ImmunoResearch Laboratories (West Grove, PA, USA) and from GE Healthcare (Baie d’Urfé, QC, Canada), respectively. Alexa Fluor 488 goat anti-rabbit IgG, Alexa Fluor 594 goat anti-mouse IgG and Alexa Fluor 594 donkey anti-rabbit IgG were from Invitrogen (Burlington, ON, Canada). α-tubulin (mouse 12G10) and LAMP1 (mouse UH1) antibodies were from the Developmental Studies Hybridoma Bank (Iowa City, IA, USA). Calreticulin antibody was from Affinity Bioreagents (Golden, CO, USA).

### 2.3. Isolation of Human PMNs

Healthy adult volunteers were recruited and they signed a consent form in accordance with the Declaration of Helsinki. The Institutional Review Board of the Centre de recherche du CHU de Québec-Université Laval approved the study (Project 2012-1030, B12-08-1038/F9-70243). Fresh human blood was collected from healthy volunteers into isocitrate anticoagulant solution. PMNs were purified as described previously [35]. PMNs were resuspended in Mg^2+^-free HBSS, pH 7.4, containing 1.6 mM CaCl_2_.

### 2.4. Electrophoresis and Immunoblotting

Samples (1.5 × 10^6^ PMNs or PLB-985 were mixed with preheated 2X Laemmli’s sample buffer (1X is 62.5 mM Tris-HCl pH 6.8, 4% SDS, 5% 2-mercaptoethanol, 8.5% glycerol, 2.5 mM Na_3_VO_4_, 10 mM pNPP, 10 µg/mL each of aprotinin and leupeptin, and 0.025% bromophenol blue) and boiled for 7 min. Proteins were separated by SDS-PAGE on 10% acrylamide or on 7.5–20% gradient acrylamide gels, and transferred to PVDF membranes. Immunoblotting was performed using the indicated antibodies and revealed with the HRP-conjugated secondary anti-rabbit or anti-mouse antibodies (1/20,000) and the Western Lightning Chemiluminescence detection system (PerkinElmer, Woodbridge, ON, Canada).

### 2.5. AGAP2 Immunoprecipitation

PMNs were centrifuged and lysed on ice in 1 mL of non-denaturing lysis buffer, 50 mM Tris-HCl pH 7.4, either 1% NP40, 10% glycerol, 137 mM NaCl, 2 mM EDTA, 10 µg/mL of each aprotinin and leupeptin, 2 mM Na_3_VO_4_, 10 mM pNPP, 1 mM PMSF, 3 mM DFP, 250 µg/mL soybean trypsin inhibitor, and briefly vortexed. After preclearing with protein A-Sepharose (GE Healthcare, Baie d’Urfé, QC, Canada) the 10,000× *g* supernatant supplemented with 0.005% BSA was incubated with rotation for 2 h at 4 °C with 50 µL of protein A-Sepharose (50% slurry) pre-coupled with 5 µL of the AGAP2 serum. The beads were washed three times with lysis buffer, mixed with 50 µL of preheated 2X Laemmli’s sample buffer (1X is 62.5 mM Tris-HCl pH 6.8, 4% SDS, 5% 2-mercaptoethanol, 8.5% glycerol, 2.5 mM Na_3_VO_4_, 10 mM pNPP, 10 µg/mL each of aprotinin and leupeptin, and 0.025% bromophenol blue) and boiled for 7 min. Samples were resolved by 7.5–20% gradient SDS-PAGE as previously described [36]. Where indicated the gel was stained with SYPRO Ruby and the band of interest analyzed using mass spectrometry as previously described [36,37]. All MS/MS samples were analyzed using Mascot (Matrix Science, London, UK; version 2.2.0). Oxidation of methionine and phospho-serine, phospho-threonine and phospho-tyrosine were specified in Mascot as variable modifications. Scaffold (version Scaffold_4.8.9, Proteome Software Inc., Portland, OR, USA) was used to validate MS/MS based peptide and protein identifications.

### 2.6. Metabolic Labelling with ^32^P

Where specified, neutrophils (5 × 10^7^ cells/mL) were incubated the cells were metabolically labeled with [^32^P]-orthophosphate by incubation with 1 mCi/mL in nominally phosphate- and calcium-free medium (20 mM HEPES, 150 mM NaCl, 5 mM KCl, and 10 mM glucose) for 1 h at 37 °C. Unincorporated radioactivity was discarded and the cells were washed twice in HBSS. The cells were resuspended at 2 × 10^7^ cells/mL in HBSS for 10 min at 37 °C. The cells (10^7^) were stimulated with opsonized Zymosan A in a ratio of 10 particles per cell or with 1.5 mg MSU crystals for the indicated times. PMNs were centrifuged and lysed on ice in 1 mL of non-denaturing lysis buffer and AGAP2 was immunoprecipitated as described above. The samples were resolved by 7.5–20% gradient SDS-PAGE, the gels were dried and apposed to X-ray films at −80 °C for up to a week.

### 2.7. Plasmid Constructs and PCR Mutagenesis

pCI-FLAG-AGAP1 (NM_014914) was a gift of Z. Nie (Medical College of Georgia, Augusta, GA, USA). AGAP2/KIAA0167 cDNA (NM_014770) was obtained from the Kazusa DNA Research Institute (Chiba, Japan). The open reading frame of AGAP2 was subcloned into the *Xh*oI/*Kpn*I sites of pAcHLT-B (BD Biosciences, Mississauga, ON, Canada), into the *BglII*/*Kpn*I sites of pEGFP-C1 (Clontech, Mountain View, CA, USA) and into the *BamH*I/*Not*I sites of pcDNA3.1/HisC (Invitrogen, Burlington, ON, Canada). Primers incorporating restriction sites (underlined) were used to generate pEGFP-C1-AGAP2 deleted of the first 329 amino acids (pEGFP-C1-AGAP2ΔGLD: Primers #2 and 3) and pEGFP-C2-AGAP2 deleted between amino acids 561 and 710 (pEGFP-C2-AGAP2ΔGAP: Primers #1, 4, 5 and 6). Primer 1, 5′-CGC-GGA-TCC-ATG-CAT-GCCCAG-AGG-CAG-TT-3′; primer 2, 5′-TGG-GGT-ACC-GCG-GCC-GCC-TAT-ACC-AGCGCA-ACC-GGG-G-3′; primer 3, 5′-CGC-GGA-TCC-ATG-CGG-GGA-GAG-ACA-ACAGGG-AG-3′; primer 4, 5′-CAC-CTC-GAG-TCT-CAC-AGC-ATT-GCA-GAC-TGG-CTA-3′; primer 5, 5′-CAC-CTC-CAG-ACG-TGG-CTA-CCG-TTC-TCC-TGC-TTT-3′; primer 6, 5′CAC-GTC-GAC-AGT-TTG-TGT-CTT-CTG-GAA-GGC-GTG-3′. AGAP2 cDNA in pEGFPC1 was used as a template for PCR. Site-directed mutagenesis of AGAP2 was performed using pcDNA3.1-HisC-AGAP2 as a template, and the QuickChange Site-Directed Mutagenesis Kit (Agilent Technologies, La Jolla, CA, USA). AGAP2 cDNAs with the K83AS84N and the R618K point mutations were sub-cloned by PCR into pEGFP-C3.

### 2.8. Expression and Purification of AGAP2

AGAP2 cDNA was inserted into the pAcHLT-B baculovirus shuttle vector and co-transfected with linearized BaculoGold viral DNA (BD Biosciences, Mississauga, ON, Canada) into Sf9 insect cells. Sf9 cells were infected with viruses at multiplicity of infection ~1. Cells were collected 48 h post-infection, and 6xHis-tagged AGAP2 was purified on a Ni-NTA Agarose column according to the manufacturer’s instructions. Protein purity was evaluated by SDS-PAGE followed by Coomassie staining.

### 2.9. Cell Culture and Transfection

PLB-985 cells (German Collection of Microorganisms and Cell Culture) were grown in RPMI 1640 medium containing 10% FBS and 2 mM L-glutamine at 37 °C, in a humidified atmosphere of 5% CO_2_. To induce a neutrophil-like phenotype, cells were cultured with 0.3 mM dbcAMP for 3 days. After one day of differentiation 2 × 10^6^ PLB-985 cells were transfected using the Nucleofector (Lonza, Walkersville, MD, USA). Cells were transfected with 3 µg of AGAP2 siRNA #1 (Hs_CENTG1_3_HP siRNA; Qiagen, Mississauga, ON, Canada) or #2 (Hs_CENTG1_4_Hp siRNA, Qiagen), or non-silencing siRNA (AllStars Negative Control siRNA, Qiagen), in 100 µL of nucleofection buffer (25 mM Hepes, 120 mM KCl, 2 mM MgCl_2_, 10 mM K2HPO_4_, 5 mM L-cysteine) using program U-002. AGAP2 siRNA #1 sequence used was: 5′-CCU-UGA-ACA-AAG-AAU-GGA-A-3′ and AGAP2 siRNA #2 sequence was: 5′GAA-UCU-UCU-UCA-ACA-GCA-A-3′. After nucleofection the cells were immediately transferred into pre-warmed complete RPMI 1640 medium containing 0.3 mM dbcAMP. Cell functions were monitored at 48 h post-transfection. Western blotting was used to assess AGAP2 expression levels, which were normalized to those of actin. CHO-IIA cells (a gift from Sergio Grinstein, The Hospital for Sick Children, Toronto, ON, Canada) stably expressing human FcγRIIA were maintained in α-MEM containing 10% FBS, 2 mM L-glutamine and 0.5 mg/mL G418. Transfection was carried out using 3 µL of the Fugene-6 Transfection Reagent (Roche Applied Sciences, Laval, QC, Canada) and 1 µg of DNA. Cells were used within 24 h of transfection. Stable transfectants expressing GFP-AGAP2 were selected and sorted using an EPICS XL flow cytometer (Beckman Coulter, Fullerton, CA, USA). RAW264.7 macrophages stably expressing mCherry actin (a gift from David A. Knecht, University of Connecticut, Storrs, CT, USA) were grown in DMEM with 5% FBS at 37 °C in 5% CO_2_ in a humidified atmosphere. Cells (2 × 10^6^) were nucleofected in solution V using 2 µg of DNA and program D-32. Phagocytosis was analyzed 6–7 h after nucleofection.

### 2.10. RT-PCR

RNA was extracted from 10^7^ PMNs or PLB-985 using Trizol reagent (Invitrogen, Burlington, ON, Canada). RNA (1 µg) was reverse-transcribed using random priming and SuperScript II Reverse Transcriptase (Invitrogen). Primers to amplify human AGAP2, AGAP1, and ribosomal protein RPLP0 were: AGAP2 #1 (467 bp product), forward, 5′-AGA-TGG-GTGAAG-GCC-TGG-AAG-CCA-C-3′, reverse, 5′-CGT-TCC-GGA-TCG-CCT-GGA-TGG-CCAC-3′; AGAP2 #2 (216 bp product), forward, 5′-ATG-CAT-GCC-CAG-AGG-CAG-TT-3′, reverse, 5′-GCG-CAG-TTC-AGG-AAT-GGA-GC-3′; AGAP1 (721 bp product), forward, 5′ACA-TCT-ACT-CCA-TCT-ACG-AGC-TGC-3′, reverse, 5′-GCT-GAT-TGT-GCA-CGGCAG-ACA-CC-3′; RPLP0 (248 bp product), foward 5′-GTT-GTA-GAT-GCT-GCC-ATT-G-3′, reverse 5′-CCA-TGT-GAA-GTC-ACT-GTG-C-3′.

### 2.11. Confocal Microscopy

Samples were imaged on an Olympus IX-70 laser-scanning confocal microscope (Olympus, Hamburg, Germany) using a 60X PlanApo/1.4 NA or a 100X UPlanApo/1.35 NA oil immersion objective. Z-series of images were collected at 0.5 µm or 1 µm increments with Olympus FluoView FV300 (version 4.3) acquisition software. Image analysis was performed using Volocity 4 (PerkinElmer, Woodbridge, ON, Canada) and ImageJ software (v1.43k) (National Institutes of Health, Bethesda, MD, USA; http//rsb.info.nih.gov/ij/ (accessed on 16 June 2011). Images shown are representative of at least three independent experiments. Single confocal cross sections of cells are represented. CHO-IIA cells were plated (2–3 × 10^5^) on glass coverslips overnight and then transiently transfected with pEGFPC1-AGAP2. To visualize microtubules cells were fixed with Cytoskelfix reagent (Cytoskeleton, Denver, CO, USA), permeabilized with 0.1% Triton X-100 in PBS, incubated with mouse anti-α-tubulin (1/25) for 1 h and after washing stained with Alexa 594-conjugated goat anti-mouse antibody (1/400) for 30 min. An identical procedure was used visualize LAMP1 in CHO-IIA cells (anti-LAMP1 undiluted). To visualize endoplasmic reticulum cells were fixed with 3% PFA for 10 min at room temperature and permeabilized with 0.1% Triton X-100 in PBS for 15 min before blocking for 30 min with 10% FBS in PBS. The anti-calreticulin antibody (1/200) was incubated with the samples for 1 h, which were then stained with Alexa 594-conjugated donkey anti-rabbit antibody (1/400) for 30 min. Cells transfected with pcDNA3.1/HisC-AGAP2 were fixed with 4% PFA, permeabilized with 0.1% Triton X-100 in PBS and incubated with our rabbit anti-AGAP2 polyclonal antibody at 1/250 for 1 h. After washing with PBS (FBS 5%, Triton X-100 0.05%), samples were stained with Alexa 488-conjugated goat anti-rabbit antibody (1:200) for 30 min. Mitochondria were stained (30 min at 37 °C) using 160 nM of MitoTracker CMXRos (Invitrogen, Burlington, ON, Canada). Filamentous actin was detected using Alexa Fluor 594-conjugated phalloidin (Invitrogen, Burlington, ON, Canada). After washing and fixation with Cytoskelfix the samples were mounted using ProlongGold antifade reagent. All procedures were performed at room temperature. To quantitatively evaluate the level of colocalization, Pearson’s correlation coefficient was calculated on background-subtracted images. The analysis was performed using an ImageJ plug-in (JaCoP) [38].

### 2.12. Phagocytosis Assays

CHO-IIA (2 × 10^5^ cells) plated on cover slips were transiently transfected with pEGFP-C1-AGAP2, pEGFP-C2-AGAP2ΔGAP, pEGFP-C1-AGAP2ΔGLD, pEGFP-C3AGAP2R618K or pEGFP-C3-AGAP2K83AS84N. 24 h post-transfection, the samples were centrifuged at 300× *g* for 1 min with human IgG-opsonized 2.93 µm beads at a bead-to-cell ratio of 15:1. The cells were then incubated for the indicated times at 37 °C. Non-internalized beads were detected by incubating the samples with Texas Red-labeled, donkey anti-human IgG for 20 min at 4 °C prior to fixation with 4% PFA. Phagocytosed beads were visualized by confocal microscopy using a 60X PlanApo oil immersion objective. The phagocytic index was defined as the number of beads ingested per transfected cell from randomly chosen fields. A minimum of 100 cells was counted per condition. To quantify GFP and actin at the phagocytic cups, CHO-IIA were allowed to phagocytose human IgG-opsonized beads for 1 min. Samples were fixed with Cytoskelfix, permeablized with 0.1% Triton X-100 and stained with Alexa Fluor 594-conjugated phalloidin. Single confocal cross sections of cells were acquired at non-saturating settings. The signal intensity on selected linear regions of each protein at the phagocytic cup was quantified using Image J Color Profiler plug-in. RAW264.7 macrophages stably expressing mCherry-actin were plated on glass coverslips overnight and overlaid with 20 µL of human IgG-opsonized latex beads (3.87 µm) in serum-free DMEM. Samples were centrifuged and incubated for 2 min at 37 °C prior to fixation with 4% PFA. PLB-985 and dbcAMP PLB-985 cells (2 × 10^6^) were resuspended in 150 µL RPMI 1640 medium containing 2 µM calcein-AM and were allowed to adhere to glass coverslips for 30 min at 37 °C in a 5% CO_2_ atmosphere. Alexa 594-conjugated zymosan A BioParticles were opsonized by incubation for 1 h at 37 °C in 50% pooled and decomplemented human serum diluted in PBS, and then added to the coverslips in a ratio of 10 particles per cell. Incubation was performed for 35 min at 37 °C, and phagocytosis was stopped by washing with ice cold PBS. Cells were then fixed in 4% PFA for 20 min. All samples were mounted using Prolong Gold antifade reagent and viewed using a 60X PlanApo oil immersion objective. Images were collected with the Olympus FluoView FV300 acquisition software and analyzed with Volocity 4 and ImageJ.

### 2.13. FACS Analysis of Phagocytosis

Alexa 488-conjugated zymosan A BioParticles were opsonized as described above. Particles were added to 25 µL of PLB-985, dbcAMP PLB-985, or siRNA dbcAMP PLB-985 cell suspensions (10^7^/mL) in RPMI 1640, in a ratio of 10 particles per cell. Synchronization was achieved by centrifuging the cells with zymosan particles and incubating the samples for the indicated time at 37 °C. Phagocytosis was halted by adding 300 µL of ice cold HBSS and keeping the samples on ice until flow cytometry analysis. Two min prior to FACS analysis, 0.4 mg/mL trypan blue in citrate buffer (pH 4.4) was added to quench the fluorescence of non-internalized particles. Samples were analyzed using an EPICS XL flow cytometer. A total of 10,000 viable cells were counted. The mean fluorescence intensity (MFI) multiplied by the percentage of viable cells that had ingested fluorescent particles was used to quantify the phagocytic response (phagocytic capacity).

### 2.14. Statistical Analysis

Statistical analyses were carried out using either the Student’s unpaired *t*-test (two-tailed) or a single factor ANOVA followed by Tukey’s multiple comparison when more than two means were considered. Values of *p* < 0.05 were deemed statistically significant. All results presented are expressed as mean values ± SEM. For all figures, an asterisk (*) denotes a *p*-value of < 0.05, whereas two asterisks (**) denote a *p*-value of < 0.01. Calculations were made with GraphPad Prism 4 software.

## 3. Results

### 3.1. AGAP2, but Not AGAP1 or PIKE-L, Is Expressed in Human Neutrophils

To identify regulators of the small GTPases Arf and characterize the ArfGAPs expressed in human polymorphonuclear neutrophils (PMNs), we assessed the expression of AGAP1 and AGAP2 via RT-PCR, using primers specific for either AGAP1 or AGAP2 [12]. As shown in Figure 1A, AGAP1 expression was undetectable in PMNs, while AGAP1 was amplified when the pCI-FLAG-AGAP1 vector construct was used as positive control. AGAP2 message was detected in PMNs using two different sets of primers including primer set #2, amplifying a N-terminal region that is unique to AGAP2 (Figure 1C [39]). Since Northern blot analyses have shown the presence of the ArfGAPs AGAP1/GGAP1 and AGAP2/GGAP2, but not of AGAP3 mRNAs in peripheral blood leucocytes, the expression of AGAP3 in PMNs was not investigated [7,8,40]. Having detected AGAP2 mRNAs, we produced a rabbit polyclonal antibody raised against amino acids 131–279 of AGAP2 to monitor its expression in PMNs. As shown in Figure 1B, Western blot analyses using a short exposure time revealed that the antibody detects as little as 18 ng of recombinant AGAP2 purified from Sf9 cells. Since the region 131–279 of AGAP2 shares some identity with that of AGAP1 (58%), the specificity of the antibody was characterized. The antibody was not specific for AGAP2 and cross-reacted with pCI-FLAGAGAP1 over-expressed in CHO-IIA cells (Figure 1C) which did not express either endogenous AGAP1 or AGAP2 [41]. We previously reported that AGAP2, but not AGAP1 or PIKE-L, is expressed by human PMNs [36]. In PMNs the AGAP2 antibody recognized a protein having the expected molecular weight of AGAP2 (~90 kDa).

### 3.2. Sub-Cellular Localization of AGAP2 in CHO-IIA Cells

Because expression of AGAP2 in PMNs was too low to assess its sub-cellular distribution by confocal microscopy following labeling with the AGAP2 antibody (not shown), we assessed the intracellular distribution of His6-tagged AGAP2 over-expressed in CHO-IIA cells. As shown in Appendix A, AGAP2 was mainly localized in the perinuclear cytoplasmic region, with reticulate networks and a punctate pattern at the cell periphery. Using the same confocal settings, the pre-immune serum failed to detect any specific signal in cells over-expressing AGAP2 (Appendix A). To assess the putative compartments to which AGAP2 localizes in CHO-IIA cells, we fused green fluorescent protein (GFP) to the *N*-terminus of AGAP2. As shown in Figure 2A, the perinuclear, reticular and punctate distribution of GFP-AGAP2 in transiently or stably transfected CHO-IIA cells was similar to that of the His6-tagged protein, suggesting that addition of a N-terminal tag has no detectable impact on AGAP2 sub-cellular localization. In contrast to GFP, no His6-tagged AGAP2 (Appendix A) or GFP-AGAP2 (Figure 2A) was detected in cell nuclei. Since many members of the ArfGAP family have been shown to impact actin cytoskeletal remodeling [42], we monitored filamentous actin (F-actin) distribution in CHO-IIA cells expressing GFP-AGAP2. GFP-AGAP2 did not co-localize with F-actin to a noticeable extent in the cytoplasmic perinuclear region but showed partial co-localization with actin stress fibers at the cell periphery (overall cell Pearson’s correlation coefficient = 0.46) (Figure 2B).

Microtubules, another component of the cytoskeleton, have been reported to co-localize with the long AGAP2 isoform (PIKE-L) in HeLa cells [43]. Labeling with a α-tubulin antibody revealed of a network of long tubular structures radiating from near the nucleus toward the cell periphery. Again, AGAP2 did not co-localize significantly with the microtubules, but it was found closely juxtaposed to them (overall cell Pearson’s correlation coefficient = 0.42) (Figure 2C). The reticulate appearance of GFP-AGAP2 in the cytoplasm may be the result of a close apposition of AGAP2 with the membrane network of the endoplasmic reticulum (ER). Staining of the ER using a calreticulin antibody (ER’s resident protein) shows minor co-localization of GFP-AGAP2 with calreticulin with a Pearson’s correlation coefficient of 0.29 (Figure 3A). AGAP2 has also been shown to localize with mitochondria in three different cell lines [44]. To explore this possibility in CHO-IIA cells, the localization of GFP-AGAP2 was compared to that of mitochondria labeled with a red fluorescent dye (MitoTracker CMXRos). Mitochondria showed minor co-localization with GFP-AGAP2 (Pearson’s correlation coefficient = 0.38) (Figure 3B). To assess whether AGAP2 localizes to known vesicular structures we also stained CHO-IIA cells with an antibody against the late endosomal and lysosomal marker LAMP1. LAMP1 showed small punctate structures slightly concentrated next to the nucleus with more diffuse labeling at the periphery of the cell co-localizing partially with GFP-AGAP2 (Pearson’s correlation coefficient = 0.43) (Figure 3C). Taken together, the data indicate that GFP-AGAP2 co-localizes, at least partially, with polymerized actin, microtubules, and the late endosomal/lysosomal compartments of unstimulated CHO-IIA cells.

### 3.3. AGAP2 Is Transiently Recruited to the Phagocytic Cup during FcγR-Mediated Phagocytosis

This section may be divided by subheadings. It should provide a concise and precise description of the experimental results, their interpretation, as well as the experimental conclusions that can be drawn. AGAP2 over-expression has been reported to disassemble focal adhesions [45]. Because several components involved in focal adhesion assembly are also important for phagocytosis [46], we assessed whether AGAP2 plays a role in FcγR-mediated phagocytosis in CHO-IIA. We chose CHO-IIA cells as a phagocytic model because they are amenable to transfection compared to the refractoriness of PMNs to genetic manipulations. Moreover, CHO-IIA cells, as shown in Figure 4A, strongly captured IgG-opsonized, but not non-opsonized, latex beads. Thus, signaling pathways that follow bead binding can be assigned directly to FcγRIIA receptors. Additionally, these cells approximate the phagocytic process [47,48] and important associated events observed only upon IgG-opsonized bead engagement such as phosphorylation of FcγRIIA receptors on tyrosine residues, as shown in Figure 4B, and as previously described in professional phagocytes such as PMNs [49]. CHO-IIA cells transiently expressing GFP or GFP-AGAP2 were challenged with human IgG-opsonized latex beads for the indicated times at 37 °C, fixed and stained with Alexa 594-labeled phalloidin to monitor actin polymerization in a collar-like structure at the phagocytic cup, an early event in FcγR-mediated phagocytosis [50]. A recruitment of AGAP2 to nascent phagosomes with enrichment of F-actin was observed in CHO-IIA cells (Figure 5A, upper row, arrows). Line-plots of pixel intensities across the phagocytic cup and other areas of the cell body revealed an approximate 2-fold enrichment of GFP-AGAP2 at the phagocytic cup in comparison with an adjacent cytoplasmic region. Maximum peak intensities of GFP-AGAP2 and Alexa 594-labeled actin were superimposed at the phagocytic cup (Figure 5A, upper row).

In contrast, GFP did not co-localize with or was not enriched at the actin-rich structures during the early stages of phagocytosis (Figure 5A, lower row, arrows). Accumulation of AGAP2 was only observed at the phagocytic cups (Figure 5B, arrows) and absent from the fully formed (sealed) phagosomes devoid of polymerized actin (Figure 5B, arrowheads). To ensure that this was not an artifact of the CHO-IIA model system, we repeated the experiments using professional phagocytes (RAW267.4 macrophages) stably transfected with mCherry-actin (Figure 5C). As observed for CHO-IIA cells, enrichment of GFP-AGAP2 and of polymerized actin was noticeable during the initial stage of phagocytosis (arrow) and absent from sealed phagosomes (arrowhead) in RAW 267.4 cells. These data suggest that AGAP2 is rapidly but transiently recruited to forming phagosomes in both CHO-IIA cells and RAW264.7 macrophages.

### 3.4. AGAP2 Increases the Phagocytic Efficiency of CHO-IIA Cells

The transient accumulation of AGAP2 at the phagosome cup combined with previous work showing inhibition of FcγR-mediated phagocytosis by the Arf-GAP ASAP2/PAG3/KIAA0400 [28], raised the hypothesis that AGAP2 regulates phagocytosis. As shown in Figure 6A (top panel), CHO-IIA cells transiently transfected with GFP or GFP-AGAP2 were capable of internalizing opsonized beads over time with a leveling off tendency at 30 min. In comparison to GFP-transfected cells, AGAP2 enhanced the number of internalized particles at all time points by an average of 17% to 42%.

The phagocytic efficiency of cells stably expressing GFP-AGAP2 was also enhanced in comparison to control CHO-IIA cells (Figure 6A, lower panel). These results suggest that AGAP2 regulates an early event in phagocytosis. AGAP2 contains a N-terminal GLD domain followed by a PH domain, the Arf GAP and the ANK repeat domains [2]. To characterize the domain(s) responsible for the AGAP2-mediated increase in phagocytosis, we generated ΔGLD and ΔGAP deletion mutants (Figure 6B, upper panel). Figure 6B shows that deletion of the GAP domain, but not of the N-terminal GLD domain, abolished the ability of AGAP2 to enhance phagocytosis of IgG-opsonized beads by CHO-IIA cells. To assess whether deletion of these domains affects the recruitment of AGAP2 to the phagocytic cup, CHO-IIA cells transfected with either GFPAGAP2 ΔGAP or GFP-AGAP2 ΔGLD were incubated with IgG-opsonized beads for 1 min at 37 °C. As shown in Figure 6C, GFP-AGAP2 ΔGAP was present throughout the cytoplasm but not enriched at the nascent phagosomes displaying actin cups. In contrast, GFP-AGAP2 with its deleted GLD domain accumulated at the phagocytic cup in a pattern similar to that of GFPAGAP2 (Figure 6D). Conversely, in cells over-expressing the GAP domain of AGAP2, large vesicular structures were observed throughout the cytoplasm but they never reached the phagocytic cups [41]. Thus, the GAP domain alone is not sufficient for localizing AGAP2 to the forming phagocytic cups.

It could be argued that deletion of domains changes the conformation of AGAP2 and interferes with its activity. To address this possibility, we generated the double mutant (K83AS84N) to create a GTPase-inactive GLD domain [13,51]. As reported above for the ΔGLD mutant, the ability of the K83AS84N mutant to promote phagocytosis of IgG-opsonized beads was similar to that of GFP-AGAP2 (Figure 7A). We also mutated in the GAP domain the arginine (R618K) that is critical for the GAP function of AGAP2 [12]. Contrary to the ΔGAP mutant, the ability of the GAP-deficient mutant (R618K) to enhance phagocytosis was similar to that of GFP-AGAP2 (Figure 7A). Of note, both GFP-AGAP2 R618K and GFP-AGAP2 K83AS84N were recruited to phagocytic cups and were found to colocalize with F-actin cups as shown for wild type AGAP2 (Figure 7B,C). These results suggest that the GAP domain of AGAP2, but not its GAP activity or the GLD domain, is required to promote FcγRIIA-mediated phagocytosis.

### 3.5. AGAP2 Silencing in the Myeloid Cell Line PLB-985 Decreases Phagocytosis

PLB-985 cells differentiated to granulocytes with dbcAMP (dibutyryl cyclic adenosine monophosphate) are suitable for the study of FcγR-mediated phagocytosis [52]. Figure 8A shows that FcγRIIA expression is not detected in undifferentiated cells whereas a treatment with 0.3 mM dbcAMP for 3 days increases FcγRIIA expression. As expected, undifferentiated PLB-985 failed to internalize opsonized zymosan particles despite their presence in their vicinity. In contrast, opsonized particles were rapidly engulfed by dbcAMP-differentiated PLB-985 cells (Figure 8B,C).

To assess the function of AGAP2 in PLB-985 cells we monitored AGAP2 expression at the mRNA and protein levels. As shown in Figure 9A, the expression of AGAP2 mRNA, but not that of AGAP1, was detected in both undifferentiated and dbcAMP-differentiated cells as we observed in human PMNs (Figure 1A). Moreover, similar levels of AGAP2 protein were detected in undifferentiated and differentiated PLB-985 (Figure 9B). Having established the expression of AGAP2 in PLB-985 cells, we next used RNA interference to silence AGAP2. The two siRNAs tested reduced the levels of endogenous AGAP2 by ~44% (siRNA #1) and ~34% (siRNA #2) in dbcAMP-differentiated PLB-985 in comparison to cells transfected with the control non-silencing siRNA (Figure 9C). Phagocytosis of opsonized zymosan particles was then monitored in dbcAMP-differentiated PLB-985 silenced for AGAP2. AGAP2-silenced cells show a small decrease in their phagocytic capacity. At 10 min the phagocytic capacity of cells treated with AGAP2 siRNA #1 and siRNA #2 was reduced by 16% (*p* = 0.07, *n* = 7) and by 23% (*p* = 0.02, *n* = 7), respectively (Figure 9D). Taken together the data indicate that AGAP2 is not essential for FcγR-mediated phagocytosis but is a positive regulator of phagocytosis in professional phagocytes.

### 3.6. Particulate Agonists Induce AGAP2 Phosphorylation in Human PMNs

Phosphorylation by various kinases regulates AGAP2 interaction with various proteins and impacts its GAP activity toward Arf proteins [53]. In the next series of experiments, we evaluated whether stimulation with the chemotactic peptide fMLF, opsonized zymosan and MSU crystals induces AGAP2 phosphorylation. Briefly, PMNs metabolically labeled with [^32^P]-orthophosphate were stimulated with, fMLF, zymosan and MSU crystals and AGAP2 immunoprecipitated under non-denaturing condition, followed by electrophoretic separation on SDS-PAGE and autoradiography. The results of these experiments are presented in Figure 10. The left panels show the kinetics of AGAP2 phosphorylation in response to opsonized zymosan. AGAP2 phosphorylation was near maximal at 2.5 min and plateaued up to 30 min, the last time tested (Figure 10A). AGAP2 was also phosphorylated in PMNs stimulated with MSU crystals. MSU crystals induced a sustained AGAP2 phosphorylation up to 15 min that declined to near basal levels by 30 min (Figure 10B). There was no significant increases in AGAP2 phosphorylation up to 5 min post-stimulation with 100 nM fMLF [41]. Analyses of AGAP2 post-translational modifications by mass spectrometry identified a phospho-peptide ^469^RAL**S**TDCTPSGDLSPLSRE^487^ only in PMNs incubated with opsonized zymosan and MSU crystals but not in non-stimulated cells (data shown). The peptide sequence encompasses the Akt phosphorylation site (Ser-472) within the PH domain of AGAP2 [53,54].

## 4. Discussion

This study provides insight into a novel function of AGAP2 in FcγR-mediated phagocytosis. Three lines of evidence suggest that AGAP2 contributes to phagocytosis. First, in CHO-IIA cells, AGAP2 localized primarily to the actin-rich phagocytic cup, an early stage of the phagocytosis process, but not at the later stages of phagosome maturation where the vast majority of the phagosomes are devoid of F-actin [55]. Second, AGAP2 enhanced FcγRIIA-mediated phagocytosis in CHO-IIA cells, and silencing AGAP2 in neutrophil-like PLB-985 cells decreased phagocytosis. Third, AGAP2 promotes phagocytosis independently of its GLD domain and of its GAP activity. A gene producing multiple protein isoforms by alternative use of exons or start codons encodes for PIKE-L, PIKE-S and AGAP2, also named PIKE-A [6] and GGAP2 [7]. AGAP2 is part of a larger family that includes AGAP1 and AGAP3 [2]. AGAP1 expression has been detected in human peripheral blood leucocytes by Northern blot analysis (Xia et al., 2003). This study shows that the expression of AGAP2, but not of AGAP1, was detected at the mRNA and protein levels in PMNs and PLB-985. The band recognized by the AGAP2 antibody, which also cross-reacted with AGAP1, was compatible with the predicted molecular weight of AGAP2. Immunoprecipitation combined with mass spectrometry confirmed the expression AGAP2 in human PMNs [36].

Actin polymerization at nascent phagosomes results from the combined action of the Rho family of small GTPases, Rac and Cdc42 [56,57,58,59,60,61]. Rab and Arf families of small GTPases are also involved in FcγR-mediated phagocytosis. Specifically, Rab11 promotes trafficking of recycling endosomes to the macrophage surface [56] and Arf6 mediates the focal delivery of vesicles, possibly from the same endocytic compartment, to the forming phagosome [23,62], and also controls cytoskeletal reorganization in early steps of phagocytosis [27,28,29]. Deletion of Arf6 in mouse PMNs reduces phagocytosis [63]. Arf1, most studied for its functions in the Golgi apparatus [64,65], also contributes to phagocytosis through recruitment of the endosome-associated clathrin adaptor complex AP-1 at early stages of phagocytosis [32]. Various GAPs regulate chemotaxis and ROS production by mouse PMNs. These include RhoGAPs [66] and ArfGAP family members GIT2 and ARAP3 [67,68]. GAPs can modulate phagocytosis through their GAP activity or their scaffolding properties. For example, the Ras GTPase Rap1GAP is rapidly activated following FcγR cross-linking and localizes near the phagosome cup in NR8383 macrophages. Not surprisingly, over-expression of active Rap1GAP severely inhibits ingestion of IgG-opsonized red blood cells [69]. Over-expression of ASAP2 (PAG3/KIAA0400), the only ArfGAP known to regulate phagocytosis, but not that of its GAP-deficient mutant, attenuated focal accumulation of F-actin beneath IgG-opsonized beads and formation of phagocytic cups possibly through inhibition of Arf6 activity [28]. Phagocytosis is also regulated by the Ras-like GAP CAPRI [70]. However, the effect of CAPRI is not linked to its Ras GAP activity but the protein functions as a scaffold to localize Rac1 and Cdc42 to phagocytic cups during FcγR-mediated phagocytosis. In this study, we established that AGAP2 functions as a positive regulator of FcγR-mediated phagocytosis and that the effect of AGAP2 is independent of its GAP function. It remains unknown how AGAP2 regulates phagocytosis but our data suggest that the protein functions as an adaptor to promote FcγR-mediated phagocytosis. Though AGAP2 has been shown to efficiently catalyze GTP hydrolysis on Arf1 and Arf5 in vitro [12,36], over-expression of AGAP2 protein had no measurable effects on the levels of Arf1-GTP in intact cells [71]. However, it cannot be excluded that AGAP2 may act on a relatively small pool of Arf1, for instance at the forming phagosome [72], that could not be detected using the Arf1-GTP pull-down assay. Interestingly, Zhu and her colleagues recently showed that while AGAP2 disrupted focal adhesion, the GAP-deficient R618K mutant was as effective as the wild-type protein in inducing focal adhesion dissolution [45]. It has also been reported that centaurin-α1, another ArfGAP, acts on stress fiber formation, cell spreading and focal adhesion independently of its GAP activity [73]. Taken together, the data suggest that AGAP2 does not act to steer the delicate balance of GTP or GDP bound to Arfs for promoting phagocytosis.

The recruitment of AGAP2 to the phagocytic cup upon ingestion of IgG-opsonized beads raises the possibility that AGAP2 acts as a modulator of FcγR-mediated phagocytosis either by acting directly on the phagocytic process or indirectly by transporting elements (vesicles, proteins) to the forming phagocytic cups. In neurons, centaurin-α1 is recruited to the plasma membrane through direct interaction of its GAP domain with the kinesin microtubule-associated motor protein KIF13B [74]. KIF13B is also known to transport phosphatidylinositol-(3,4,5)-triphosphate-containing vesicles along microtubules plus ends through centaurin-α1 [75,76]. The observation of microtubules ending at the location of forming phagosomes during FcγR-mediated phagocytosis in RAW264.7 macrophages [77] led to the discovery of Kif5B as the molecular motor responsible for the delivery of Rab11-containing membranes and receptor recycling for pseudopod extension and particle binding [78]. Since AGAP2 was found closely juxtaposed to microtubules in CHO-IIA cells an interaction with a kinesin motor cannot be excluded. However, this putative interaction is unlikely to be strictly dependent on the GAP domain of AGAP2, which when over-expressed in CHO-IIA cells was not recruited to the phagocytic cup but instead was mis-targeted to large intravesicular structures dispersed throughout the cytoplasm [41]. Taken together the data suggest the GAP domain is not sufficient for targeting AGAP2 to the phagocytic cup. Several reports suggest that the GLD domain binds GTP or other nucleotide triphosphates and that intra-molecular interactions with the GAP domain enhance its atypical GTPase activity [6,7,79,80,81].

In the present study, we show that the GLD domain and GTPase deficient mutants neither abrogate the transient recruitment of AGAP2 to the forming phagocytic cup nor promote FcγR-mediated phagocytosis in CHO-IIA cells. Taken together the data suggest that the GLD domain, or its GTPase activity, plays no significant role in localizing AGAP2 to the actin-rich phagocytic cup. Localized and oriented actin polymerization provides the driving force for pseudopod extension over the target particle, while insertion by focal exocytosis of endomembrane into the base of the phagocytic cup compensates for membrane loss or accommodates the extension of the plasma membrane when phagocytes engulf large or multiple particles [82,83,84]. Published data support a role for AGAP2 in membrane trafficking by affecting the intracellular distribution of the endosome-associated clathrin adaptor complex AP-1 and by acting on the AP-1/Rab4 endosomal compartment containing transferrin receptors [12,85]. Interestingly, the PH domain of AGAP2 is a protein-binding module for the clathrin adaptor protein AP-1 as well as focal adhesion kinase (FAK) [12,45]. PH domains also bind phosphoinositides, which are required for signaling and are rapidly depleted from the phagocytic cup as the phagosome seals [86,87,88]. Whether the transient recruitment of AGAP2 to the forming phagocytic cup depends on the phosphoinositide binding or scaffolding properties of the PH domain awaits further investigation.

PMN stimulation with particulate agonists resulted in a rapid and sustain phosphorylation of AGAP2. Phospho-peptide analyses identified a potential Akt phosphorylation site within the PH domain of AGAP2, Ser-472 [53,54]. Disruption of lipids raft with 10 mM methyl-β-cyclodextrin, a cholesterol-depleting agent, inhibits phagocytosis of opsonized-zymosan [89]. Like the FcγRIIA receptor, AGAP2 is almost totally soluble in 1% NP-40 in unstimulated human neutrophils [36,63,89]. Reports highlight the re-localization of FcγRIIA in high-density detergent-resistant membranes following receptor cross-linking [34,89,90]. Furthermore, MSU crystals induce the re-localization of class Ia PI-3K into high-density detergent-resistant membranes that form a complex with the protein tyrosine kinase Syk [91]. Future studies should investigate whether cross-linking of FcγR or stimulation with MSU crystals induces AGAP2 recruitment into detergent-resistant membranes and its subsequent phosphorylation.

Phosphatdylinositol-3-kinases link cell surface receptor activation to Akt signaling pathway. Activation of FcγRs in mouse or FcγRIIA in human PMNs induces the phosphorylation and the activation of Akt [92,93]. It was described previously that MSU crystals induce Akt phosphorylation downstream of phosphatdylinositol-3-kinases and Syk in human PMNs [91]. AGAP2 directly binds Akt and enhances its kinase activity [39]. In cancer cells, this positive feedback activation loop between AGAP2 and Akt enhances AGAP2 phosphorylation and promotes its association with UNC5B, thereby inhibiting the UNC5B-dependent programmed cell death [54]. Future studies should determine whether Akt associates and phosphorylated AGAP2 and if AGAP2 deletion accelerates PMN apoptosis.

## 5. Conclusions

In summary, we show in the present study that professional phagocytes, polymorphonuclear neutrophils, express AGAP2. AGAP2 participates in the very earliest steps of phagocytosis. The protein was transiently recruited to the actin-rich phagocytic cup, an early stage of the phagocytosis process, but absent at the later stages of phagosome maturation where the majority of the phagosomes were devoid of polymerized actin. Though AGAP2 is not essential for phagocytosis, it increases the engulfment of IgG-opsonized particles by phagocytes. The GTPase and GAP activities of AGAP2 were not required to promote FcγR-mediated phagocytosis. Furthermore, in human neutrophils, opsonized zymosan or monosodium urate crystals induced AGAP2 phosphorylation. We propose that AGAP2 acts as a scaffold to regulate the dynamics of phagocytosis through the recruitment of intra-vesicular compartments or proteins to the phagocytic cup.

## Figures and Tables

**Figure 1 cells-12-00072-f001:**
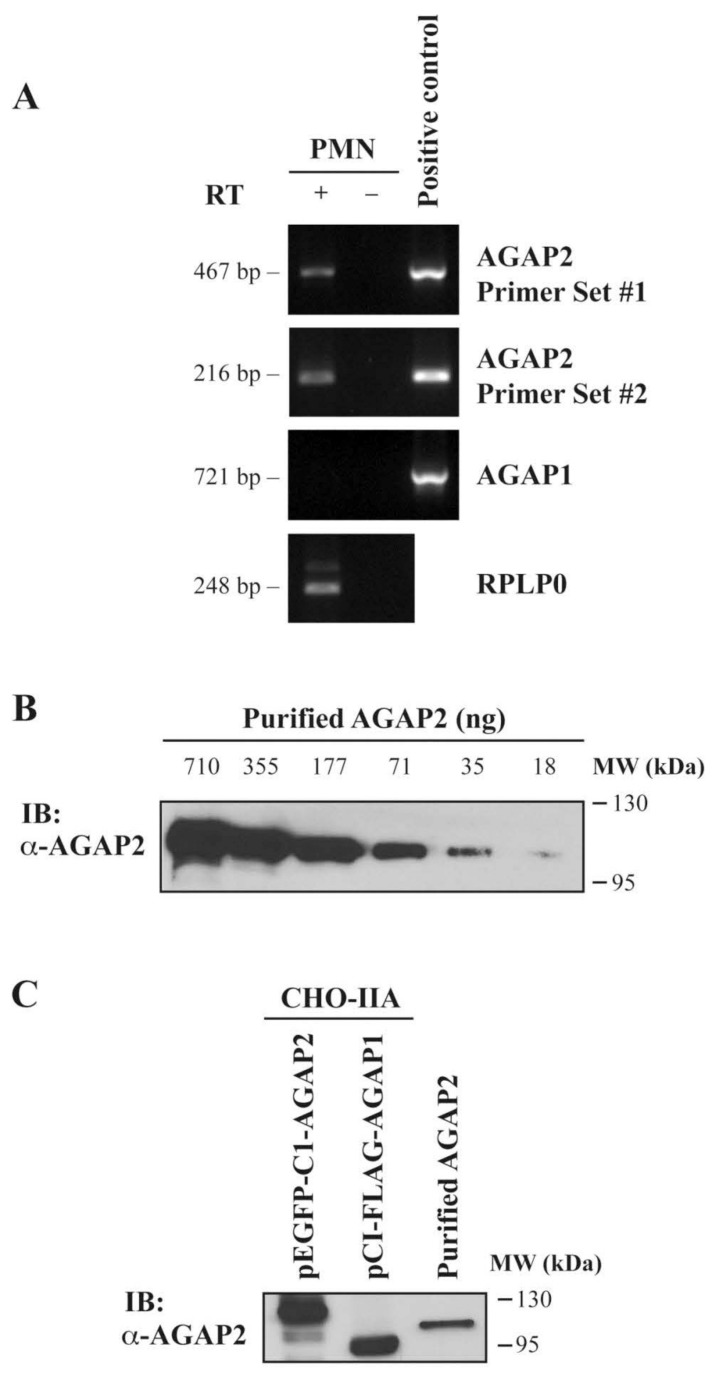
Expression of AGAP1 and AGAP2 mRNAs in human PMNs. (**A**) PCR analyses of AGAP2 (primer sets #1 and #2), AGAP1, and RPLPO expression, as control reaction, in PMNs. Reactions were performed on reverse transcription products (RT +), or on reverse transcription reactions performed without reverse transcriptase (RT −) as genomic DNA control. pEGFP-C1AGAP2 and pCI-FLAG-AGAP1 templates were used as positive controls. (**B**) Sensitivity of the polyclonal anti-AGAP2 antibody was determined using decreasing amounts of purified His6tagged AGAP2. (**C**) CHO-IIA cells were transfected with pEGFP-C1-AGAP2 or pCI-FLAGAGAP1 and specificity of the anti-AGAP2 antibody (1/5000) was examined by Western blotting. Purified His6-tagged AGAP2 was used as a control.

**Figure 2 cells-12-00072-f002:**
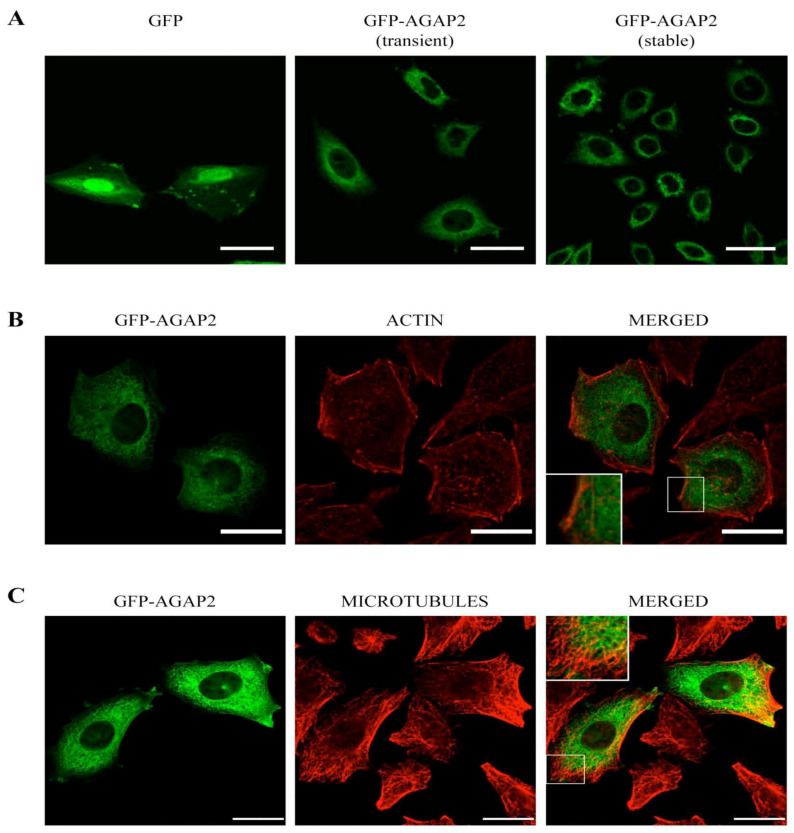
Co-localization of pEGFP-C1-AGAP2 with cytoskeletal proteins. (**A**) CHO-IIA cells transiently transfected with pEGFP-C1-AGAP2 or pEGFP-C1 alone and CHO-IIA cells stably expressing pEGFP-C1-AGAP2 were examined by confocal microscopy. CHO-IIA cells transiently expressing pEGFP-C1-AGAP2 were stained for filamentous actin with Alexa Fluor 594-conjugated phalloidin (**B**) or microtubules with anti-α-tubulin antibody (**C**). A single confocal cross section of cells is represented. Insets in (**B**,**C**) show a magnification of the boxed areas. Bars, 20 µm.

**Figure 3 cells-12-00072-f003:**
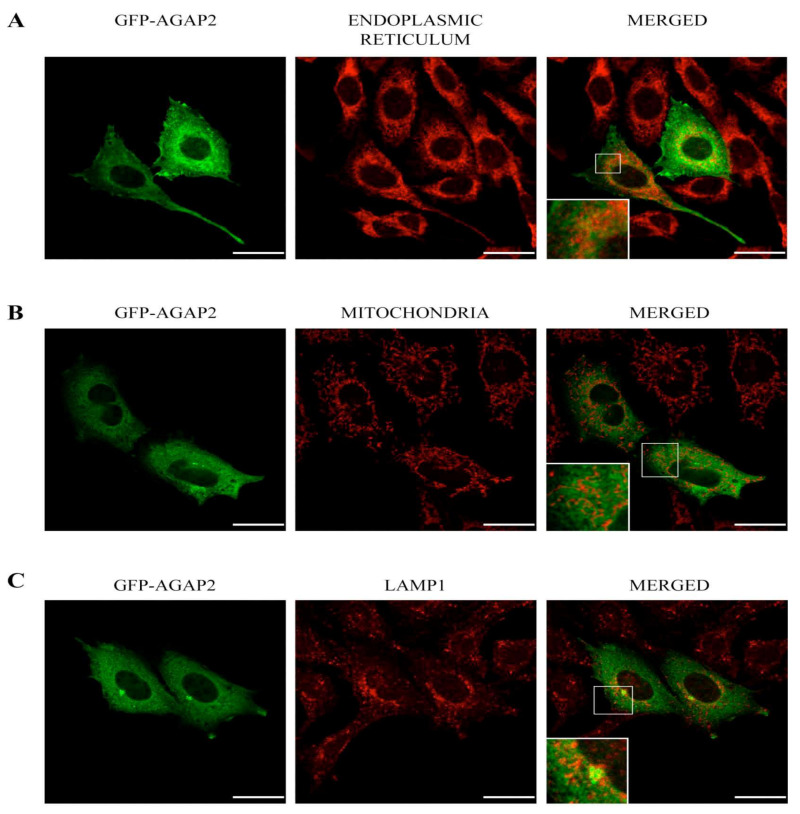
Co-localization of pEGFP-C1-AGAP2 with intracellular compartments. CHO-IIA cells transiently expressing pEGFP-C1-AGAP2 were labeled for endoplasmic reticulum with anti-calreticulin antibody (**A**), mitochondria with MitoTracker CMXRos (**B**) and late endosome/lysosome compartment with anti-LAMP1 antibody (**C**). A single confocal cross section of cells is represented. Insets show a magnification of the boxed areas. Bars, 20 µm.

**Figure 4 cells-12-00072-f004:**
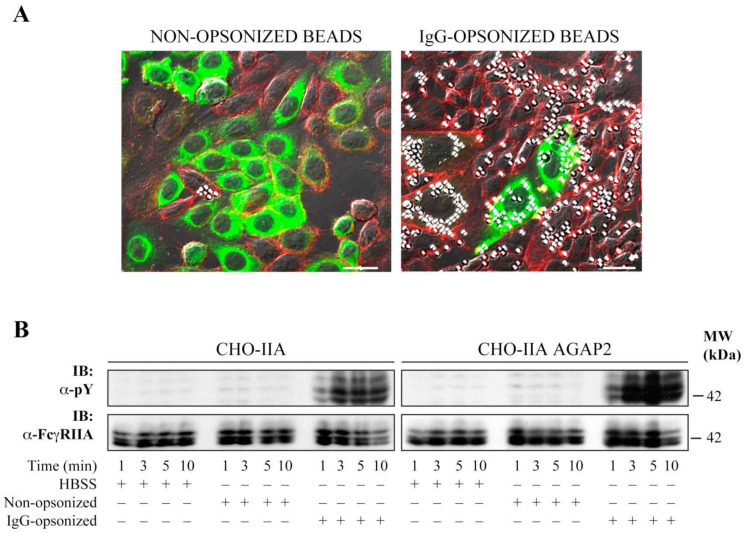
Phagocytosis of IgG-opsonized beads by CHO-IIA cells. (**A**) CHO-IIA cells transfected with pEGFP-C1-AGAP2 were incubated with non-opsonized (left panel) or IgG-opsonized (right panel) polystyrene beads for 30 min at 37 °C at a bead-to-cell ratio of 30:1. Samples were then fixed and stained for F-actin with Alexa Fluor 594-conjugated phalloidin. Cells were analyzed by confocal microscopy. Bar, 20 µm. (**B**) CHO-IIA cells (CHO-IIA) or CHO-IIA cells stably transfected with pEGFP-C1-AGAP2 (CHO-IIA AGAP2) were incubated with IgG-opsonized or non-opsonized polystyrene beads (bead-to-cell ratio of 30:1) for the indicated times. Samples were resolved by SDS–PAGE and probed with an anti-pY (4G10) (1/4000) or an anti-FcγRIIA (CT10) antibody (1/5000).

**Figure 5 cells-12-00072-f005:**
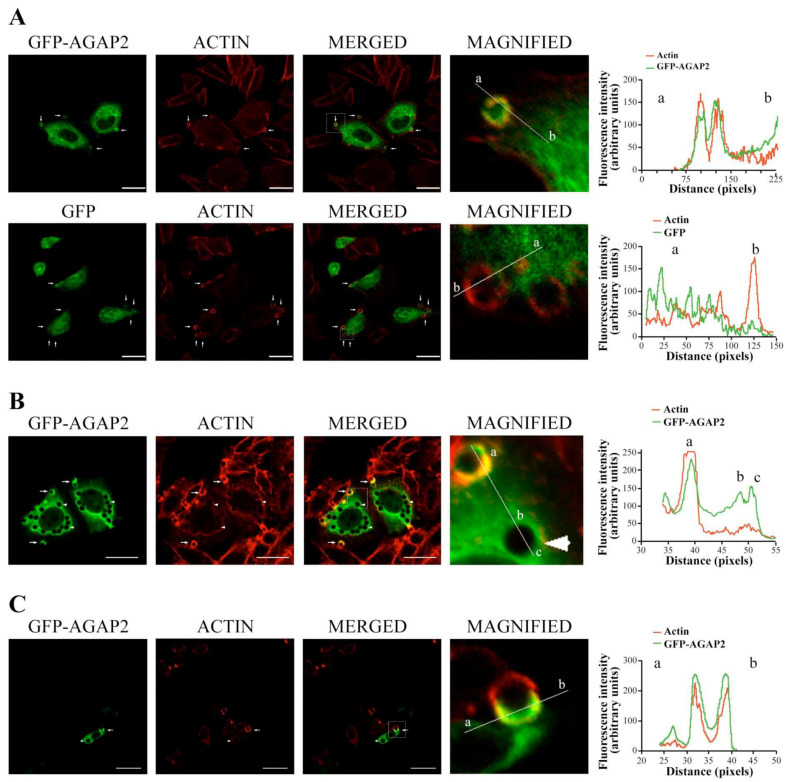
Recruitment of AGAP2 to phagocytic cups. (**A**) CHO-IIA cells transfected with either pEGFP-C1-AGAP2 (upper row) or pEGFP-C1 alone (lower row) were incubated for 1 min at 37 °C with IgG-opsonized beads and stained with Alexa 594-phalloidin (red). Confocal images were acquired. (**B**) CHO-IIA cells transfected with pEGFP-C1-AGAP2 were allowed to undergo phagocytosis for 30 min. Polymerized actin was labeled with Alexa 594-phalloidin (red). (**C**) RAW264.7 macrophages stably expressing mCherry-actin (red) and transfected with pEGFP-C1AGAP2 were allowed to internalize 3.87 µm latex beads for 2 min. Arrows and arrowheads show phagocytic cups and sealed phagosomes, respectively. Magnified images of phagocytic cups and/or sealed phagosomes from a frame indicated in each overlay panel are shown. The intensity of each fluorescent signal on a solid line is shown in the graphs. Bars, 20 µm. (**a**–**c**) Reference points of the image processing.

**Figure 6 cells-12-00072-f006:**
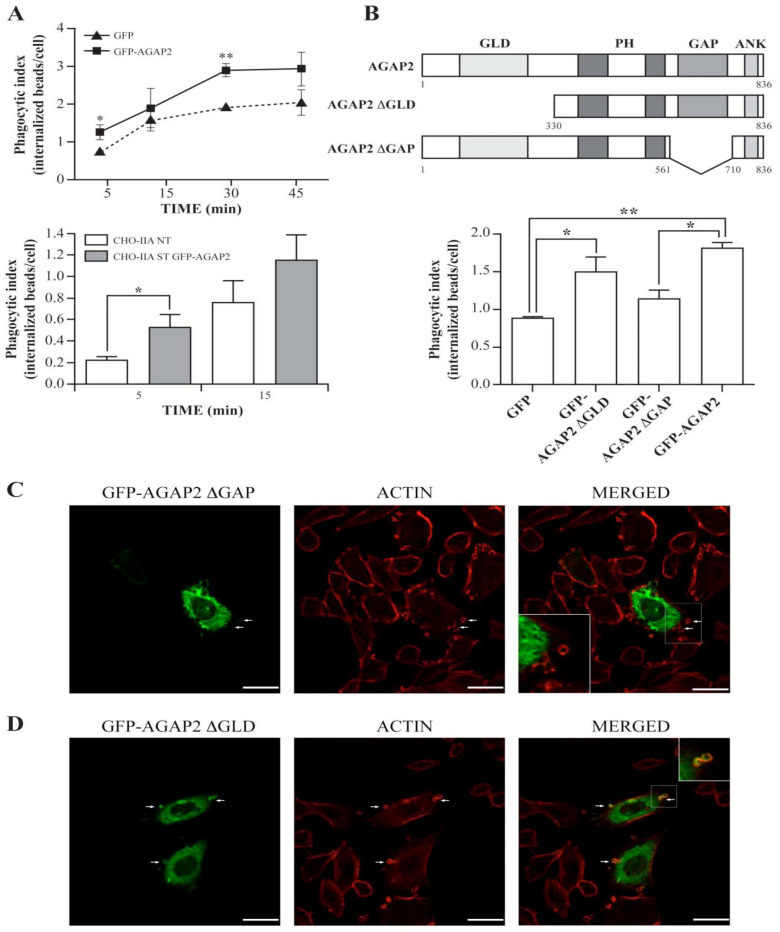
Impact of pEGFP-C1-AGAP2 on Fcγ-mediated phagocytosis. (**A**) CHO-IIA cells transiently (upper panel) or stably (lower panel, CHO-IIA ST GFP-AGAP2) expressing pEGFPC1-AGAP2 were incubated with IgG-opsonized latex beads (bead-to-cell ratio of 15:1) for the indicated times. External beads were distinguished from phagocytosed beads by staining with Texas Red-labeled donkey anti-human IgG antibody. Non-transfected CHO-IIA (CHO-IIA NT) cells and/or CHO-IIA transiently expressing GFP were used as controls. The phagocytic index was calculated as described in Section 2. The mean ± SEM of three independent experiments (two for stably transfected cells) is plotted with at least 100 cells counted in each case. (**B**) Schematic of AGAP2 and its deletion mutants. GLD, GTP-binding protein-like domain; PH, pleckstrin homology domain; GAP, GTPase-activating protein domain; ANK, ankyrin repeats (upper panel). CHO-IIA transiently expressing pEGFP-C1, pEGFP-C1AGAP2ΔGLD, pEGFP-C2-AGAP2ΔGAP or pEGFP-C1-AGAP2 were allowed to phagocytose IgG-opsonized beads for 5 min. The samples were processed as in A. The mean ± SEM of three independent experiments is plotted with at least 100 cells counted in each case (lower panel). CHO-IIA cells transiently transfected either with pEGFP-C2-AGAP2ΔGAP (**C**) or pEGFP-C1AGAP2ΔGLD (**D**) were allowed to undergo phagocytosis for 1 min. Polymerized actin was labeled with Alexa 594-phalloidin (red). Insets in (**C**,**D**) show a magnification of the boxed areas. Arrows indicate examples of F-actin phagocytic cups. Bars, 20 µm. (**A**,**B**) Data are presented as mean ± SEM with *p*-values from the Student’s unpaired *t*-test (two-tailed). An asterisk (*) denotes a *p*-value < 0.05 and two asterisks (**) denote a *p*-value < 0.01.

**Figure 7 cells-12-00072-f007:**
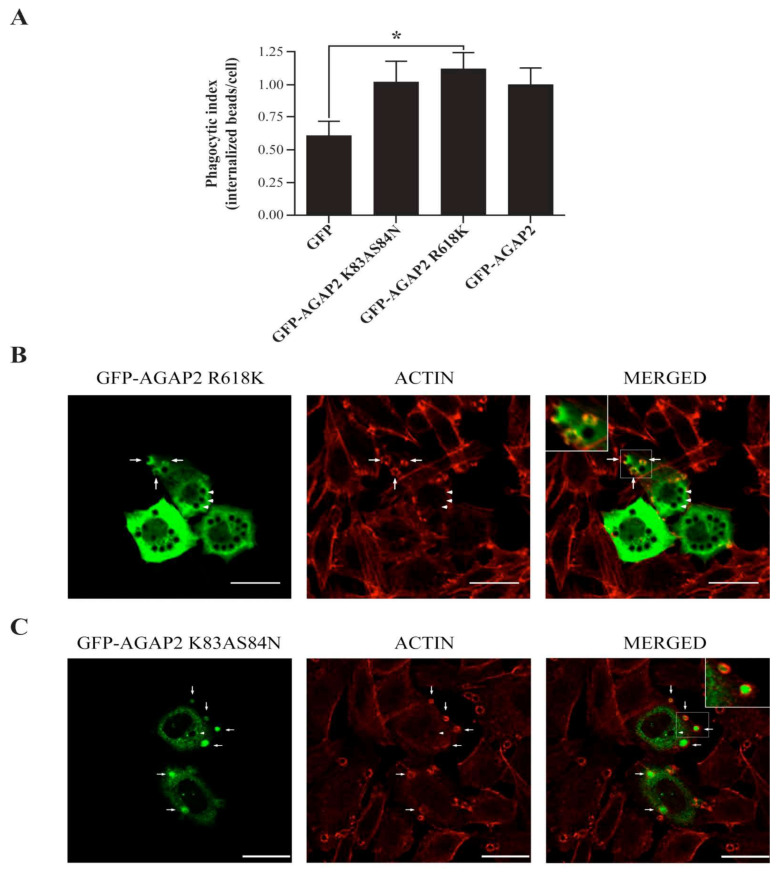
Impact of GAP-and GTPase-deficient AGAP2 on Fcγ-mediated phagocytosis. (**A**) CHO-IIA cells transiently expressing pEGFP-C1, pEGFP-C3-AGAP2K83AS84N, pEGFP-C3AGAP2R618K or pEGFP-C1-AGAP2 were allowed to phagocytose IgG-opsonized beads for 5 min. The number of internalized beads was counted and the phagocytic index calculated as described in Materials and Methods. The mean ± SEM of five independent experiments is plotted with at least 100 cells counted in each case. CHO-IIA cells transiently expressing pEGFP-C3-AGAP2R618K (**B**) or pEGFP-C3-AGAP2K83AS84N (**C**) were allowed to undergo phagocytosis for 30 min or 2 min, respectively. Polymerized actin was labeled with Alexa 594-phalloidin (red). Insets in B and C show a magnification of the boxed areas. Arrows and arrowheads show phagocytic cups and sealed phagosomes, respectively. Bars, 20 µm. (**A**) Data are presented as mean ± SEM with *p*-values from the Student’s unpaired *t*-test (two-tailed). An asterisk (*) denotes a *p*-value < 0.05.

**Figure 8 cells-12-00072-f008:**
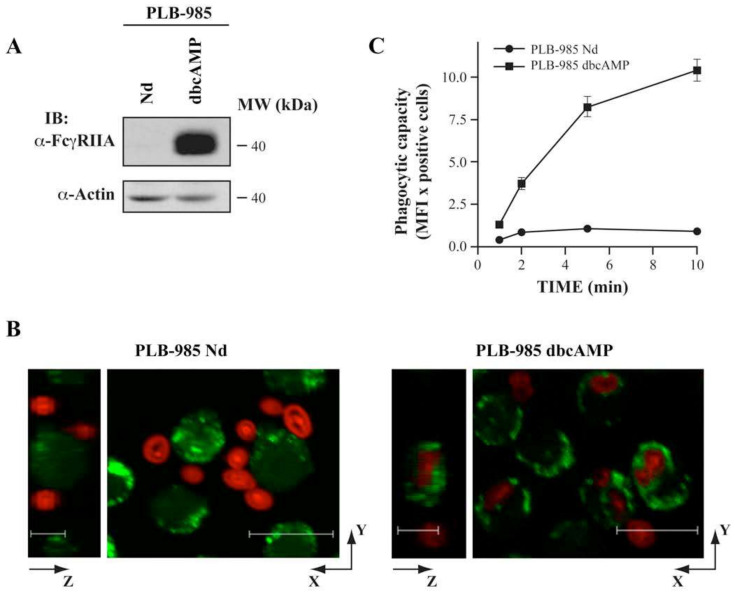
dbcAMP PLB-985 cells as a phagocytic model. (**A**) Protein samples (30 µg per lane) of undifferentiated (Nd) and dbcAMP-differentiated PLB-985 cells were resolved by SDS-PAGE (7.5–20% gradients gels). FcγRIIA and actin were detected by Western blotting using CT10 (1/5000) and anti-actin (1/1000) antibodies, respectively. (**B**) Confocal scanning laser microscopy of undifferentiated (Nd) and dbcAMP differentiated-PLB-985 cells phagocytosing Alexa 594-conjugated opsonized zymosan particles (red) in a ratio of 10 particles per cell. Cells labeled with Calcein-AM (green) were allowed to phagocytose particles for 35 min. Vertical sections (y versus z) are shown in the left panel, and horizontal sections (x versus y) in the right panel. Images are representative of three independent experiments. Bars, 12 and 6 µm. (**C**) FACS analysis of phagocytic capacity of undifferentiated (Nd) and dbcAMP-differentiated PLB-985. Cells were incubated with Alexa 488-conjugated opsonised zymosan particles for the indicated times. Phagocytic capacity was determined by multiplying the number of phagocytosing cells by the Mean Fluorescence Intensity (MFI). Each value represents the mean value ± SEM of four experiments, each performed in duplicate.

**Figure 9 cells-12-00072-f009:**
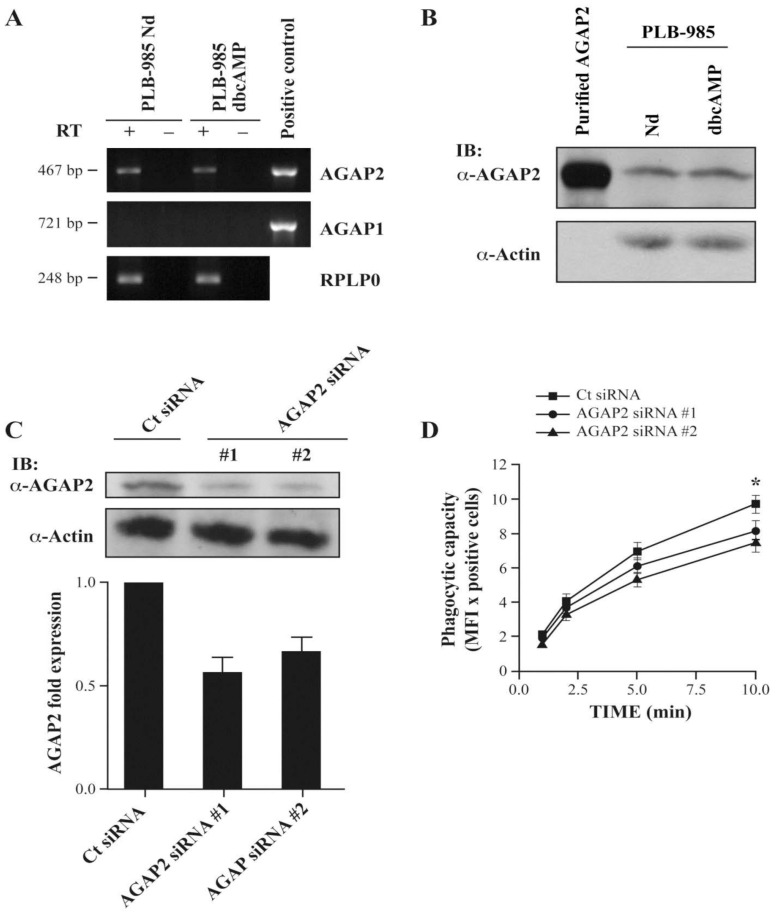
Impact of AGAP2 siRNA on Fcγ-mediated phagocytosis in dbcAMP-differentiated PLB-985 cells. (**A**) PCR analyses of AGAP1, AGAP2, and RPLP0, as control reaction, showing mRNA expression in undifferentiated (Nd) and dbcAMP-differentiated PLB-985. Reactions were performed on reverse transcription products (RT +), or on reverse transcription reactions performed without reverse transcriptase (RT −) as genomic DNA control. pEGFP-C1-AGAP2 and pCI-FLAG-AGAP1 templates were used as positive controls. (**B**) Cell lysates (80 µg protein/well) from undifferentiated (Nd) and differentiated PLB-985 were resolved by SDS-PAGE. AGAP2 and actin were detected by Western blotting using AGAP2 (1/5000) and actin (1/1000) antibodies. His6-tagged AGAP2 was used as a control. (**C**) AGAP2 silencing using siRNAs #1 and #2 and control siRNA (Ct siRNA). AGAP2 expression was analyzed 48 h post-transfection by immunoblotting (upper panel) and normalized densitometric readings of AGAP2 relative to actin were calculated (lower panel, *n* = 3). (**D**) Differentiated PLB-985 cells silenced for AGAP2 were allowed to phagocytose Alexa 488-conjugated opsonized zymosan particles as indicated in Section 2. Phagocytic capacity was determined by multiplying the number of phagocytosing cells by the Mean Fluorescence Intensity (MFI). Each value represents the mean value ± SEM of 7 experiments, each performed in duplicate. (**D**) Data are presented as mean ± SEM with *p*-values from the Student’s unpaired *t*-test (two-tailed). An asterisk (*) denotes a *p*-value < 0.05.

**Figure 10 cells-12-00072-f010:**
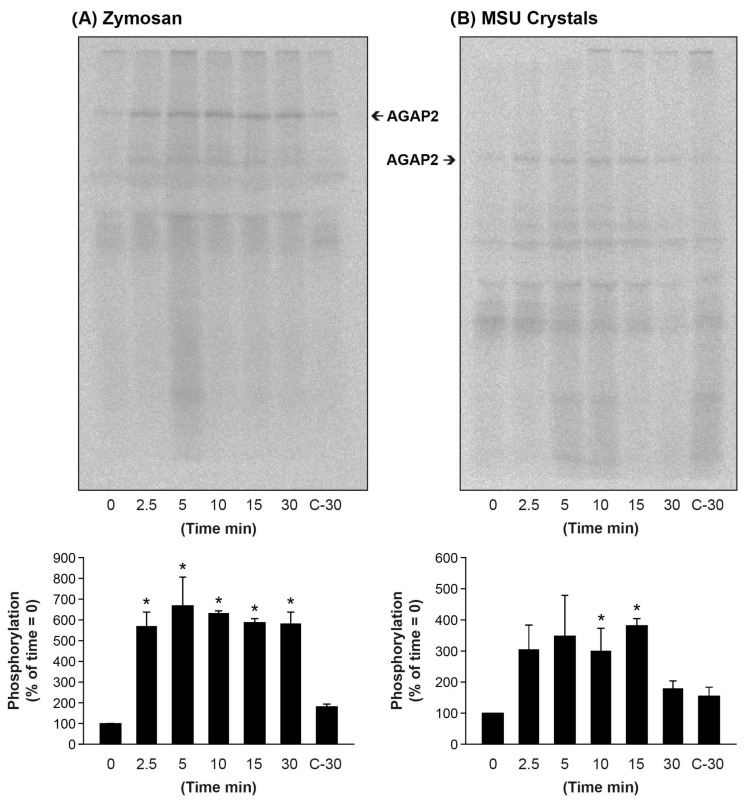
Opsonized zymosan and MSU crystals induce AGAP2 phosphorylation. ^32^P-labeled PMMs were stimulated with opsonized zymosan particles in a ratio of 10 particles per cell (**A**) or with 1.5 mg MSU crystals (**B**) for indicated times. AGAP2 was immunoprecipitated and subjected to SDS-PAGE as described in Section 2. The upper panels are one representative autoradiographic film images from 3 (**A**) and 4 (**B**) independent experiments. The films were scanned into a computed for visual and semi-quantitative densitometry (lower panels). The basal value (time 0 min) was set to an arbitrary level of 100. Data are the mean value ± SEM of 3 (**A**) and 4 (**B**) experiments, respectively. C-30 = Control 30 min. (**A**,**B**) The *p*-values were measured using single factor ANOVA followed by Tukey’s multiple comparison. An asterisk (*) denotes a *p*-value < 0.05.

## Data Availability

The data presented in this study are available on request from the corresponding author.

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
