# Peer review of "Functional Role of AGAP2/PIKE-A in Fcγ Receptor-Mediated Phagocytosis"

_cells, 2022, doi:10.3390/cells12010072_

Round 1

Reviewer 1 Report

The submitted manuscript describes a role for AGAP2, an Arf GTPase activating protein (ArfGAP) that regulates endosomal trafficking and focal adhesion remodeling, in Fcγ receptor-mediated phagocytosis. The authors used two different cell culture models, as well as primary human neutrophils, to provide evidence. The aims of the paper are clear, the methods are well described, statistical methods appear appropriate, experimental results are consistent with evidence provided and conclusions appear to follow logically from the results, and are not overstated. The manuscript is extremely well written. I only have one minor comment below:

It is not clear to this leader whether the sentence at lines 388-390 was included as an accident or on purpose. “This section may be divided by subheadings. It should provide a concise and precise description of the experimental results, their interpretation, as well as the experimental conclusions that can be drawn.” It did somewhat disrupt the flow of reading this very well written manuscript.

Author Response

Thank you for the comment.  The reviewer is right.  We forgot to remove the sentence in lines 388-390, which is part of the cell template provided by the journal. We deleted this sentence from de revised manuscript.

Reviewer 2 Report

The authors in this research article entitled “Functional role of AGAP2/PIKE-A in Fcγ-receptor-mediated phagocytosis” focus on a novel function of AGAP2 in FcγR-mediated phagocytosis. The present manuscript is engaging, and is of potential interest to the readers of Cells. I recommend the publication of this work only after the authors have addressed the following major comments:

- Since lipid rafts are critical for signaling and phagocytosis processes, the authors should investigate whether AGAP2 is associated via cholesterol-rich lipid raft to the phagocytic cup.

-The co-distribution of the two probes in fluorescence microscope images should be evaluated with an efficient metric such as the Manders Overlap Coefficient (MOC) which is implemented in image analysis software.

- Concerning the statistical analyses the authors should check that in several legends no indications of which statistical test they used for the experiments are done.

- In the figure 1 the authors should provide the densitometric analysis.

Author Response

Point 1: Since lipid rafts are critical for signaling and phagocytosis processes, the authors should investigate whether AGAP2 is associated via cholesterol-rich lipid raft to the phagocytic cup.

Response 1: Thank you for this suggestion. This will be really interesting to investigate. We have not investigated whether AGAP2 associates with the phagocytic cup through recruitment in cholesterol-rich lipid raft. We made several attempts, but low expression of AGAP2 in human neutrophils precludes using an immuno-fluorescence approach for assessing its recruitment to the phagocytic cup. We believe that this question is beyond the scope of the paper, and this is something that we would like to investigate in human neutrophils.

We know that disruption of lipids raft with 10 mM methyl-b-cyclodextrin, a cholesterol-depleting agent, inhibits phagocytosis of opsonized-zymosan [1]. Like the FcγRIIA receptor, AGAP2 is almost totally soluble in 1% NP-40 in human neutrophils [1,2]. Reports highlight the re-localization of FcγRIIA in high-density detergent-resistant membranes following receptor cross-linking [1,3,4]. Furthermore, MSU crystals induce the re-localization of class Ia PI-3K into high-density detergent-resistant membranes that form a complex with the protein tyrosine kinase Syk [5]. We agree that future studies should investigate whether cross-linking of FcγR or stimulation with MSU crystals induces AGAP2 recruitment into detergent-resistant membranes and its subsequent phosphorylation. We implemented the discussion to highlight this point (pages 692-701).

Point 2: The co-distribution of the two probes in fluorescence microscope images should be evaluated with an efficient metric such as the Manders Overlap Coefficient (-MOC) which is implemented in image analysis software.

Response 2: Thank you for your comment. The researchers can use several tools to analyze protein colocalization following the merge of fluorescent images.  The most used tools are the Pearson correlation coefficient and Mander’s overlap coefficient (MOC). Though mathematically similar, the correlation measures rely either on the absolute intensity (MOC) or the deviation from the mean (Pearson correlation coefficient). Both methods have strengths and weaknesses [6-9]. We used the JACoP plugin implemented in ImageJ to measure protein colocalization by correlation. The Pearson correlation coefficient is a well-established method for evaluating protein colocalization. Overall, Pearson correlation coefficients equal to or less than 0.5 were calculated for AGAP2 in control cells, a value that does not allow drawing strong conclusions. Relevant consideration should be given to protein localization/colocalization changes following the addition of opsonized particles to cells.

Point 3: Concerning the statistical analyses the authors should check that in several legends no indications of which statistical test they used for the experiments are done.

Response 3: Thank you for your remark. The specific statistical analysis used has been added to the figure legend as suggested.

Point 4. In the figure 1, the authors should provide the densitometric analysis.

Response 4: The data shown in Figure 1 are to validate the specificity and sensitivity of the polyclonal antibodies. Experiments were performed using recombinant AGAP2 or protein samples from cells expressing tagged-AGAP proteins in cells. Densitometric data from immunoblots are semi-quantitative.  However, densitometric analysis can provide useful information if a control for protein loading can be used for normalizing the data. No control for protein loading was performed making it impossible to provide reliable densitometric data. 

References:

  1. Marois, L.; Pare, G.; Vaillancourt, M.; Rollet-Labelle, E.; Naccache, P.H. Fc gammaRIIIb triggers raft-dependent calcium influx in IgG-mediated responses in human neutrophils. J Biol Chem 2011, 286, 3509-3519, doi:10.1074/jbc.M110.169516.
  2. Gamara, J.; Chouinard, F.; Davis, L.; Aoudjit, F.; Bourgoin, S.G. Regulators and Effectors of Arf GTPases in Neutrophils. J Immunol Res 2015, 2015, 235170, doi:10.1155/2015/235170.
  3. Rollet-Labelle, E.; Marois, S.; Barbeau, K.; Malawista, S.E.; Naccache, P.H. Recruitment of the cross-linked opsonic receptor CD32A (FcgammaRIIA) to high-density detergent-resistant membrane domains in human neutrophils. Biochem J 2004, 381, 919-928, doi:10.1042/BJ20031808.
  4. Marois, L.; Vaillancourt, M.; Marois, S.; Proulx, S.; Pare, G.; Rollet-Labelle, E.; Naccache, P.H. The ubiquitin ligase c-Cbl down-regulates FcgammaRIIa activation in human neutrophils. J Immunol 2009, 182, 2374-2384, doi:10.4049/jimmunol.0801420.
  5. Popa-Nita, O.; Rollet-Labelle, E.; Thibault, N.; Gilbert, C.; Bourgoin, S.G.; Naccache, P.H. Crystal-induced neutrophil activation. IX. Syk-dependent activation of class Ia phosphatidylinositol 3-kinase. J Leukoc Biol 2007, 82, 763-773, doi:10.1189/jlb.0307174.
  6. Dunn, K.W.; Kamocka, M.M.; McDonald, J.H. A practical guide to evaluating colocalization in biological microscopy. Am J Physiol Cell Physiol 2011, 300, C723-742, doi:10.1152/ajpcell.00462.2010.
  7. Aaron, J.S.; Taylor, A.B.; Chew, T.L. The Pearson's correlation coefficient is not a universally superior colocalization metric. Response to 'Quantifying colocalization: the MOC is a hybrid coefficient - an uninformative mix of co-occurrence and correlation'. J Cell Sci 2019, 132, doi:10.1242/jcs.227074.
  8. Adler, J.; Parmryd, I. Quantifying colocalization by correlation: the Pearson correlation coefficient is superior to the Mander's overlap coefficient. Cytometry A 2010, 77, 733-742, doi:10.1002/cyto.a.20896.
  9. Adler, J.; Parmryd, I. Quantifying colocalization: the MOC is a hybrid coefficient - an uninformative mix of co-occurrence and correlation. J Cell Sci 2019, 132, doi:10.1242/jcs.222455.

Round 2

Reviewer 2 Report

I accept in present form